# Evaluation of the Humoral Response after Immunization with a Chimeric Subunit Vaccine against Shiga Toxin-Producing *Escherichia coli* in Pregnant Sows and Their Offspring

**DOI:** 10.3390/vaccines12070726

**Published:** 2024-06-29

**Authors:** Roberto M. Vidal, David A. Montero, Adriana Bentancor, Carolina Arellano, Alhejandra Alvarez, Cecilia Cundon, Ximena Blanco Crivelli, Felipe Del Canto, Juan C. Salazar, Angel A. Oñate

**Affiliations:** 1Programa de Microbiología y Micología, Instituto de Ciencias Biomédicas, Facultad de Medicina, Universidad de Chile, Santiago 8380453, Chile; caroarellanocabezas@gmail.com (C.A.); alhecag@gmail.com (A.A.); felipedelcanto@uchile.cl (F.D.C.); jcsalazar@u.uchile.cl (J.C.S.); 2Instituto Milenio de Inmunología e Inmunoterapia, Facultad de Medicina, Universidad de Chile, Santiago 8380453, Chile; 3Centro Integrativo de Biología y Química Aplicada (CIBQA), Universidad Bernardo O’Higgins, Santiago 8320000, Chile; 4 Universidad de Buenos Aires, Facultad de Ciencias Veterinarias, Instituto de Investigaciones en Epidemiología Veterinaria, Cátedra de Microbiología, Buenos Aires C1427CWO, Argentina; aben@fvet.uba.ar (A.B.); ccundon@fvet.uba.ar (C.C.);; 5Departamento de Microbiología, Facultad de Ciencias Biológicas, Universidad de Concepción, Concepción 4070386, Chile; aonate@udec.cl

**Keywords:** Shiga toxin-producing *Escherichia coli*, STEC, chimeric protein based-vaccine, edema disease of swine, pigs

## Abstract

Shiga toxin-producing *Escherichia coli* (STEC) poses a significant public health risk due to its zoonotic potential and association with severe human diseases, such as hemorrhagic colitis and hemolytic uremic syndrome. Ruminants are recognized as primary reservoirs for STEC, but swine also contribute to the epidemiology of this pathogen, highlighting the need for effective prevention strategies across species. Notably, a subgroup of STEC that produces Shiga toxin type 2e (Stx2e) causes edema disease (ED) in newborn piglets, economically affecting pig production. This study evaluates the immunogenicity of a chimeric protein-based vaccine candidate against STEC in pregnant sows and the subsequent transfer of immunity to their offspring. This vaccine candidate, which includes chimeric proteins displaying selected epitopes from the proteins Cah, OmpT, and Hes, was previously proven to be immunogenic in pregnant cows. Our analysis revealed a broad diversity of STEC serotypes within swine populations, with the *cah* and *ompT* genes being prevalent, validating them as suitable antigens for vaccine development. Although the *hes* gene was detected less frequently, the presence of at least one of these three genes in a significant proportion of STEC suggests the potential of this vaccine to target a wide range of strains. The vaccination of pregnant sows led to an increase in specific IgG and IgA antibodies against the chimeric proteins, indicating successful immunization. Additionally, our results demonstrated the effective passive transfer of maternal antibodies to piglets, providing them with immediate, albeit temporary, humoral immunity against STEC. These humoral responses demonstrate the immunogenicity of the vaccine candidate and are preliminary indicators of its potential efficacy. However, further research is needed to conclusively evaluate its impact on STEC colonization and shedding. This study highlights the potential of maternal vaccination to protect piglets from ED and contributes to the development of vaccination strategies to reduce the prevalence of STEC in various animal reservoirs.

## 1. Introduction

Shiga toxin-producing *Escherichia coli* (STEC) can cause severe human diseases, such as hemorrhagic colitis and hemolytic uremic syndrome. Ruminants, particularly cows, are recognized as primary reservoirs for STEC, though a broad spectrum of farm animals can also harbor and shed this pathogen [1].

Notably, healthy pigs have been identified as carriers of STEC, with studies demonstrating the isolation of these bacteria from fecal samples [2]. Additionally, STEC can grow in ground pork at temperatures between 10 °C and 40 °C; therefore, pork-derived products should not be kept unrefrigerated for extended periods [3]. Although transmission of STEC from pigs to humans is relatively rare, there have been outbreaks linked to the consumption of contaminated pork products with STEC serotypes such as O157:H7, O157:NM, and O111:H- [4,5]. Recent analyses categorize STEC strains isolated from retail pork in the United States as posing low to moderate risks to human health [6]. This scenario highlights the risks of zoonoses associated with the consumption of pork products contaminated with STEC, indicating the need for thorough cooking and the implementation of comprehensive surveillance and control strategies [1,6,7,8].

The pathogenicity of STEC is primarily mediated by Shiga toxins (Stx), which belong to the AB5 family of toxins. These potent cytotoxins consist of an A subunit responsible for inhibiting protein synthesis, leading to cell death, and a pentamer of B subunits, which mediate the endocytosis of the toxin through specific interactions with cellular receptors [9]. Stx exhibits considerable diversity, with two principal types, Stx1 and Stx2, further divided into several subtypes (Stx1a, c, d, and Stx2a-o). The variation in amino acid sequences among these Stx subtypes and their specific affinity for cellular receptors influences their toxicity levels across different hosts [10,11]. Notably, Stx2 subtypes associated with human diseases exhibit a preference for the globotriaosylceramide (Gb3) receptor, whereas Stx2e, which is linked to porcine diseases, shows a higher affinity for the globotetraosylceramide (Gb4) receptor. Of note, the Gb4 receptor is prevalent in the gastrointestinal and vascular tissues of pigs, which correlates with the high virulence of Stx2e-producing STEC strains in these animals, particularly in cases of edema disease (ED) [12]. This specificity also explains why Stx2e-producing STEC strains are infrequently linked to severe disease in humans [12,13,14,15,16].

In addition to Stx2e, the F18ab fimbria constitutes another significant virulence determinant of STEC, as it enhances bacterial adherence to porcine small intestinal epithelial cells [17,18]. Interestingly, the diversity of fimbrial adhesins found in *E. coli* and other enterobacteria has a significant impact on determining the specific hosts and tissues they can colonize and infect [19,20].

ED is an infection with a worldwide distribution, primarily affecting piglets within the first two weeks post-weaning. Characterized by a high mortality rate, the clinical presentation of this disease includes symptoms such as eyelid edema, altered vocalizations, and neurological alterations, indicative of its systemic impact in affected swine [12,18,21]. ED can also occur in young fattening pigs and has been documented in wild boars, suggesting a broad ecological and epidemiological significance [21,22]. The economic impacts of ED are considerable, encompassing direct losses from mortality and morbidity, as well as indirect costs related to prevention, control measures, and potential trade restrictions due to public health concerns [23].

A unique feature in swine reproduction is the epitheliochorial placenta, which restricts maternal antibody transfer to the fetus, leaving neonate piglets reliant on lactogenic immunity [24]. Consequently, it has been suggested that the delay in the onset of ED until after weaning is due to the decline of maternal immunity during the lactation period. Other factors that could influence the delay in the onset of this disease are the absence of receptors in the intestinal epithelium of newborn pigs for F18ab fimbriae and the low expression of these fimbriae after infection, which leads to low bacterial colonization [18,25].

Traditional therapeutic and prophylactic approaches for ED face some limitations. Firstly, the efficacy of antibiotics is limited because the Stx2e has already been transported to the bloodstream when clinical signs become evident [26]. Importantly, certain antibiotics can induce the SOS response in STEC, increasing the transcription of *stx* genes and potentially exacerbating the disease [27]. Second, the growing concern over antimicrobial resistance and stricter health regulations have further limited the use of antibiotics in animal production. A prophylactic strategy has been the addition of zinc oxide (ZnO) in pig feed, which has shown some effectiveness in controlling *E. coli* infections. However, this practice also faces restrictions, especially in the European Union, due to concerns over antimicrobial resistance and the environmental impacts of ZnO [28].

These limitations have led to an increased interest in vaccines as a safer and more effective alternative to control the prevalence of STEC and ED in pigs. Thus, vaccination of pregnant sows has emerged as a promising strategy to protect neonatal piglets, taking advantage of the transfer of maternal antibodies through colostrum [29]. Licensed vaccines to prevent ED have focused on Stx2e as the main target. These vaccines, generally recombinant toxoids, are effective in inducing immunity against the toxin but do not significantly reduce STEC shedding in feces [30].

In previous research, we developed a STEC vaccine candidate based on chimeric proteins containing epitopes from Cah, OmpT, and Hes proteins. This innovative approach yielded a vaccine that not only conferred protection in a mouse model of STEC infection, but also demonstrated immunogenicity in pregnant cows during field trials [31,32]. Consequently, we are currently interested in exploring the potential use of this vaccine candidate in other STEC reservoirs.

This study aimed to evaluate the immunogenicity of our vaccine candidate in pregnant sows and the transfer of immunity to piglets. The proteins Cah, OmpT, and Hes, are present in a wide range of STEC strains isolated from both humans and cows [33,34,35]. Here we showed that the genes encoding these proteins are also widespread in STEC strains isolated from pigs. Furthermore, our results showed that this vaccine candidate elicited a robust specific IgG and IgA antibody response, which was effectively transferred to piglets, supporting the potential use of the vaccine in this species. This work provides insights into developing effective vaccination strategies to mitigate STEC prevalence and enhance food safety.

## 2. Materials and Methods

### 2.1. In Silico Detection of Virulence Genes

Genes encoding Hes (NCBI nucleotide accession CP015244, region 4,111,631–4,112,380), OmpT (CP034384.1, region 2,678,549–2,679,502), Stx2e (JQ812056.1), FedA (major structural subunit of the F18ab fimbria, KM260189), and DNA sequences encoding the passenger domains of Cah (CP040107.1, region c3,367,003–3,368,299) and Antigen 43 (Agn43, CP032667.1, region c1,811,680–1,810,274) were screened in genomes of Stx2e-producing *E. coli* and other *E. coli* strains isolated from pigs. This screening was conducted using the large-scale blast score ratio (LS-BSR) software with the blastn option [36]. Registries with BSR ≥ 0.8 were considered positives for each gene. Accession codes for genomes included in this screening are shown in Appendix A.

The Hes, OmpT, and Cah proteins are the primary targets of our STEC vaccine candidate, which was developed in previous studies [31,32]. We included the *stx2e* gene to confirm that the STEC strains analyzed were pathogenic to swine. Additionally, the *agn43* gene was included to ensure the specific detection of the *cah* gene rather than its homologs. Given the importance of the F18ab fimbria in existing vaccines against STEC in pigs, we included the *fedA* gene to compare the protective coverage provided by our target proteins against established vaccine antigens.

### 2.2. Animals

A total of 9 Landrace pregnant sows confirmed via real-time ultrasound imaging were housed in the pig production unit of the Faculty of Veterinary Sciences, at the Universidad de Buenos Aires. Sows were in pens with cement floors containing individual gestation cages, independent feeders, and waterers. The sows were transferred to individual farrowing pens in the last week of pregnancy. The piglets (n = 45) remained in pens with their mothers at the end of the trial, 30 days post-partum.

Before the first immunization, fecal samples were collected and subjected to PCR analysis using the previously described protocol for detecting STEC [37], and the herd was found to be positive.

### 2.3. Immunization Protocol

Chimera 1 (Chi1) and Chimera 2 (Chi2) proteins were produced in a veterinary pharmaceutical laboratory in accordance with good manufacturing practices, as previously described [32]. The vaccine formulations contained either 80 µg (40 µg Chi1 + 40 µg Chi2) or 200 µg (100 µg Chi1 + 100 µg Chi2) of the chimeric proteins, combined with the Montanide^TM^ Gel adjuvant. The placebo consisted of the Montanide^TM^ Gel adjuvant in phosphate saline buffer (PBS; 1X, pH 7.2).

The animals were identified with ear tags and randomly assigned to receive either a vaccine formulation or placebo following a three-dose regimen (days 56, 71, and 86 of pregnancy), which were administered via intramuscular in cervical muscle groups, behind the ear and before the angle of the shoulder. Two groups of animals (P4 and P5; n = 3 per group) received the vaccine formulations in two different doses, and a third group received only a placebo as the control group (P6; n = 3) (Figure 1).

The assessment of adverse reactions involved an examination specifically focusing on local manifestations occurring at the injection site, encompassing phenomena such as inflammation, the development of abscesses, abscess formation with subsequent ulceration, or systemic hypersensitivity. All animals maintained good health throughout the study, with no observable abnormalities at the injection sites or clinically significant differences between the groups. Furthermore, the piglets were born at term, healthy, and of normal weight, indicating that the vaccine was safe and well tolerated.

### 2.4. Sample Collection

Blood samples collected into non-anticoagulant tubes were taken from pregnant sows and piglets (5 mL) to monitor their serological profiles. Pregnant sows were sampled one day before each immunization (D-55, D-70, D-85 of pregnancy), at a week pre-partum (D-105), and on the first day post-partum. Also, two blood samples were taken from five piglets for each mother until the end of the trial, at ages D-15 and D-30 (Figure 1).

Blood samples were centrifuged at 1000× *g* for 10 min, and the supernatants were collected and stored at −20 °C until use. At the same time, fecal samples from each animal were collected via rectal swab and analyzed via PCR for the detection of STEC by using the protocol described previously [37].

### 2.5. Humoral Immune Responses

The humoral immune responses elicited by the immunizations were assessed via indirect ELISA. In short, a mixture comprising 1 µg of the chimeric proteins (Chi1 and Chi2) in 100 µL of PBS (1X, pH 7.2) was coated onto 96-well ELISA plates (Nunc-Immuno Plates, ThermoFisher, Waltham, MA, USA) and incubated overnight at 4 °C. Each well was then washed three times with 400 µL of PBS containing 0.05% Tween 20 (T-PBS). This was followed by blocking with 300 µL per well of blocking solution (T-PBS + 0.5% bovine serum albumin) for 15 min at room temperature. To measure IgG, sera were diluted from 1:250 to 1:16,000; for IgA measurements, dilutions ranged from 1: 50 to 1:1600 in the blocking solution (100 µL/well) and incubated for 60 min at 37 °C. Following incubation, wells were washed five times with T-PBS. Rabbit anti-Pig IgG (H+L) Secondary Antibody, HRP (Cat # PA1-84625, Invitrogen, Waltham, MA, USA) or Goat anti-Pig IgA Secondary Antibody, HRP (Cat # PA1-84625, Invitrogen, USA), both diluted 1:1000 in blocking solution, were added (100 µL/well) and incubated for another 60 min at 37 °C. Following five washes with T-PBS, the plates were developed with 3,3′,5,5′-tetramethylbenzidine (TMB) liquid substrate (T0440, Sigma-Aldrich, St. Louis, MO, USA) for 10 min at room temperature. The absorbance was read at 405 nm using the Synergy HT microplate reader (Biotek Instruments, Winooski, VT, USA). Each sample was analyzed in duplicate and with at least three independent replicates.

### 2.6. Statistical Analysis

Statistical differences in the distribution of virulence genes between groups of strains were assessed using contingency tables and Fisher’s exact test (Appendix A). When any of the cell values of the contingency table were zero, a value of 0.5 was added to all cells (Haldane correction) to avoid errors in the statistical test. Statistical differences in anti-Chi1/Chi2 IgG and IgA antibody levels were determined using the Kruskal–Wallis test, followed by Dunn’s multiple comparison test. A *p* value of <0.05 was considered significant for all statistical tests.

## 3. Results

### 3.1. Distribution of Cah, Ompt, and Hes Genes among STEC Strains Isolated from Pigs

The design of effective vaccines depends on targeting antigens that are conserved and prevalent within the pathogen of interest [38]. In alignment with this principle, our vaccine candidate against STEC targets the immunogenic proteins Cah, OmpT, and Hes, which have been identified in a wide range of STEC strains from humans and cows [33,34,35,39,40].

In light of this, we initially determined the distribution and frequency of the *cah*, *ompT*, and *hes* genes within *E. coli* strains isolated from pigs. A search of the GenBank database yielded 773 genomes, including 172 STEC strains and 601 strains of other *E. coli* types (Appendix A). These strains were isolated from 19 countries and belong to 288 serotypes. Among the STEC strains, the most prevalent serotypes were O155:H21 (16/178; 9%), O139:H1 (14/178; 7.9%), O100:H30 (13/178; 7.3%), O9:H30 (9/178; 5.1%), and O8:H9 (9/178; 5.1%). For non-STEC *E. coli*, the most common serotypes were O101:H9 (23/597; 3.9%), O9a:H4 (17/597; 2.8%), O101:H10 (17/597; 2.8%), O9a:H30 (14/597; 2.3%), and O9a:H9 (14/597; 2.3%). This result revealed the wide diversity of *E. coli* strains circulating in swine populations worldwide. Critically, this diversity highlighted the need for vaccines targeting STEC-specific antigens, which are prevalent in a wide range of serotypes of this pathogen.

Subsequently, we conducted a large-scale blast score ratio (LS-BSR) analysis to determine the frequency of our target antigens within the genome datasets. This analysis revealed that the *cah* (60/172; 34%) and *ompT* (82/172; 47%) genes were significantly more frequent in STEC strains compared to other types of *E. coli* isolated from swine (Table 1). In contrast, the *hes* gene was found in only a limited number of STEC strains (3/172; 1.7%). Remarkably, at least one of the three target genes was identified in 63.5% (106/172) of STEC strains, a significantly higher proportion than the 25.5% (152/601) observed in other types of *E. coli*. Additionally, the *agn43* gene, a homolog of the *cah* gene, was found at a similar frequency in both STEC (27/172; 15.6%) and other *E. coli* strains (74/601; 12.3%), indicating its wide distribution in this bacterium. Of note, the combined frequency of *cah*, *ompT,* and *hes* genes in STEC (106/172; 63.5%) was higher than that of the *fedA* gene (46/172; 26%), which encodes the major subunit of F18 fimbriae. These findings support the potential of the *cah* and *ompT* genes as targets for developing a vaccine against STEC in swine.

### 3.2. Humoral Response in Pregnant Sows

Serum levels of anti-Chi1/Chi2 IgG and IgA antibodies were determined using ELISA before the first immunization and following each dose of the vaccine. Initial serological assessments revealed that pre-immune sera from all sows were seropositive for Chi1 and Ch2 proteins, establishing a baseline of specific antibodies that that was consistent across all groups (Figure 2A). This suggests prior exposure to STEC among the animals.

After the first dose, the P5 group exhibited an increase in specific IgG levels compared to pre-immune levels and those observed in the P4 and P6 groups (Figure 2A). In contrast, a substantial increase in specific IgG levels within the P4 group was only observed after the administration of two doses. Following the second dose, the P4 and P5 groups showed elevated specific IgG levels compared to the P6 control group.

Similarly, an increase in specific IgA levels was observed in the P5 group after the first dose compared to the other groups (Figure 2B). The P4 group also showed an increase in specific IgA levels after receiving two doses of the vaccine. Intriguingly, an unexpected rise in specific IgA levels was observed in sow No. 121 of the P6 control group after the second dose. Nonetheless, an increase in specific IgA levels was clearly evident in both the P4 and P5 groups relative to pre-immune levels. These findings indicated that the vaccine candidate can induce specific IgG and IgA responses in serum.

### 3.3. Passive Transfer of Maternal Anti-Chi1/Chi2 Antibodies to Piglets

To evaluate the maternal antibody transfer from vaccinated sows to their offspring, we determined specific antibody levels in serum samples from piglets at 15 and 30 days post-birth. The initial analysis at 15 days post-birth included ten piglets from each vaccinated groups (P4 and P5) and eight from the control group (P6). At day 30 post-birth, the same cohorts were reassessed; however, the sample size for the P6 group was reduced to five piglets due to logistical constraints.

The results indicated that piglets born and nursed by sows in the P4 and P5 groups exhibited high levels of specific IgG and IgA antibodies on day 15, maintaining similar levels until day 30 post-birth, albeit with a slight reduction (Figure 3). In contrast, piglets from the P6 group showed only basal levels of these specific antibodies. This finding demonstrated that the maternal anti-Chi1/Chi2 antibodies elicited by the vaccine were successfully transferred to the piglets through colostrum and milk consumption, providing them with early humoral immunity.

## 4. Discussion

Colostrum deprivation in piglets increases susceptibility to STEC infection, highlighting the crucial role of lactogenic immunity in protecting newborn piglets against ED [41]. Indeed, several studies advocate for maternal vaccination as a preventive strategy against this and other diseases affecting piglets [29,42,43]. Moreover, reducing the zoonotic risk associated with STEC in swine populations would significantly improve public health by enhancing food safety.

This research focused on evaluating the immunogenicity of a chimeric protein-based vaccine candidate against STEC in pregnant sows and the subsequent transfer of immunity to their offspring. Given the zoonotic and economic concerns related to STEC infection and carriage in pigs, the diversity of serotypes found in our genomic analysis demonstrates the need for a vaccine that covers a wide range of strains. The *cah* and *ompT* genes were significantly more frequent in STEC strains than other types of *E. coli*, supporting their selection as antigens for a STEC vaccine in pigs (Table 1). Most compelling was the observation that the combined frequency of the *cah*, *ompT,* and *hes* genes was higher than that of the *fedA* (63% vs. 26%) gene, a critical virulence factor in porcine pathogenic STEC strains [17,18]. These findings support the potential of this vaccine candidate to target a wide range of STEC strains in swine.

Vaccines targeting F18ab fimbriae decrease STEC colonization in pigs and thus reduce the amount of Stx2e entering the systemic circulation [18,30,44]. The F18-fimbriae gene cluster includes genes such as *fedA* (coding for the major structural subunit), *fedB* (coding for the molecular chaperone), *fedC* (coding for the introducer protein), *fedE* (coding for the minor structural subunit), and *fedF* (coding for the adhesin) [45]. In particular, the *fedF* gene is highly conserved and strains lacking it exhibit markedly reduced adhesion capabilities; therefore, it has been identified as a primary target for vaccine development [46]. However, our genomic analysis indicated that a sizable proportion of Stx2e-producing STEC strains lacked this fimbria, potentially allowing them to evade immunity conferred by fimbriae-targeted vaccines.

Additionally, STEC vaccines targeting Stx2e are designed to alleviate the severity of ED by eliciting neutralizing anti-Stx2e antibodies [30]. It is crucial to acknowledge that although these anti-Stx antibodies can mitigate the intensity of the disease, they do not affect the colonization and shedding of STEC.

A limitation of our study was the uncertain role of the Cah and OmpT proteins in STEC colonization and infection in pigs. However, all sows were already seropositive for the Chi1 and Chi2 proteins before immunization, indicating prior exposure to STEC. This suggest that the bacterium expresses these antigens during colonization of the pig intestine. This observation also underscores the importance of considering background immunity when evaluating vaccine efficacy.

In ungulates, including cattle and swine, IgG constitutes for over 75% of all immunoglobulins in colostrum, emphasizing its pivotal role in neonatal immunity [47]. Furthermore, the epitheliochorial placenta in sows restricts the transfer of maternal antibodies during gestation, rendering piglets dependent on lactogenic immunity for early protection [24]. In particular, neonatal piglets absorb these colostral antibodies, primarily IgG, through the gut in a non-selective manner during the first hours of life. However, this ability to absorb immunoglobulins significantly diminishes between 18 and 36 h after birth, a period known as “gut closure” [48,49].

Additionally, functionally intact IgG within the intestinal lumen can bind antigens, contributing to the immune exclusion [47,50,51,52,53,54]. This dual function of IgG, providing both systemic and mucosal immunity, underscores the importance of developing vaccines capable of inducing responses from both IgG and IgA.

In the context of the immune responses elicited by our vaccine candidate, it is important to note that IgG was the predominant antibody type generated, with smaller amounts of IgA (Figure 2). Notably, the P5 group showed a faster humoral response after the first dose, which could be related to a higher concentration of the chimeric proteins. The slight increase in specific IgA levels in the control group after the second dose might suggest non-specific immune responses or exposure to STEC during the study. Although no STEC was isolated from feces, initial PCR results suggested the presence of this bacterium in the herd.

This vaccine candidate increased the levels of specific IgG in the serum of pregnant sows and, consequently, in the colostrum, leveraging the natural physiological mechanisms of immune protection in ungulates. An interesting finding was the effective passive transfer of specific antibodies from immunized sows to their offspring, with antibody levels remaining relatively high from days 15 to 30 post-birth. However, a slight decrease reflected a progressive decline in maternal protection as the piglets aged. The long-term persistence of these specific antibodies in piglets is uncertain, and further active immunization may be necessary.

Although the serum levels of IgA in vaccinated sows were not notably high compared to those in control sows (Figure 2), it is noteworthy that piglets nursed by immunized sows exhibited a significant increase in serum IgA (Figure 3). This result indicates that, along with IgG, IgA was also effectively transferred passively to the piglets.

While this study provides preliminary data on the immunogenicity of our vaccine candidate in pigs, it did not include a challenge assay to assess protection against infection directly. Future research will include challenge assays to evaluate the effect of the vaccine on STEC colonization and shedding in pigs. Additionally, we will explore the correlation between serum antibody levels in sows and their piglets to better understand the transfer and efficacy of induced immunity.

Together with previous studies in cattle [32], these findings support the possibility of a trans-species vaccination platform that could mitigate the impact of STEC in animal reservoirs. Implementation of such vaccination strategies could improve existing management practices, contributing to a safer food supply chain and reducing the risk of foodborne outbreaks.

## Figures and Tables

**Figure 1 vaccines-12-00726-f001:**
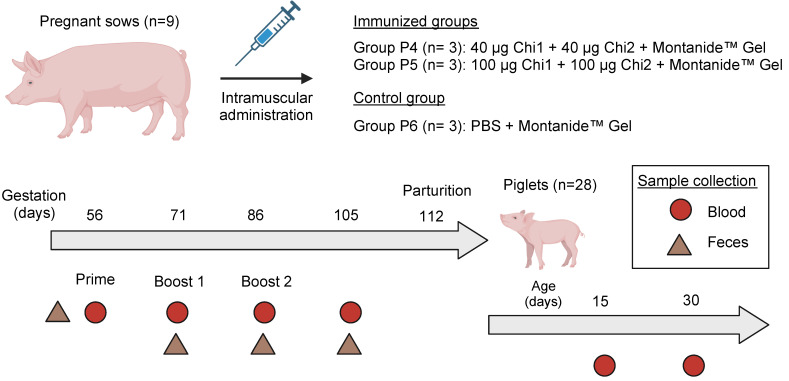
Overview of the experimental design used in this study. This figure presents the experimental design utilized to evaluate the immunogenicity of a chimeric protein-based vaccine against STEC in pregnant sows and the subsequent transfer of passive immunity to their offspring. The study involved nine pregnant sows, divided into two immunized and one control group. The immunization schedule consisted of the intramuscular administration of the vaccine, with the prime dose administered in the 8th week of gestation, followed by two booster doses at the 10th and 12th weeks. After parturition, 28 of the 45 piglets from the vaccinated and control groups were selected to assess lactogenic immunity. Sample collection for analysis included blood and feces, as detailed in the key legend.

**Figure 2 vaccines-12-00726-f002:**
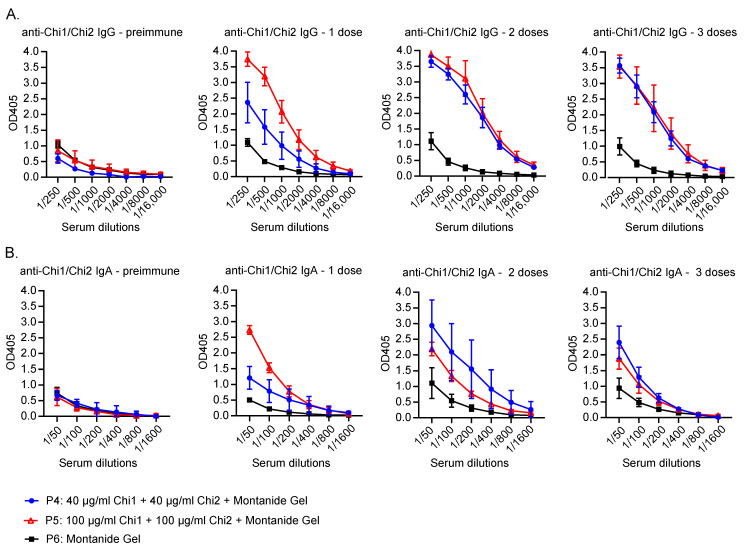
Serum antibody response in immunized swine. Sera samples were analyzed in duplicate, and the results are presented as means ± SEM of absorbance values at 405 nm for each serum dilution, n = 3 animals per group. (**A**) Anti-Chi1/Chi2 IgG antibodies. (**B**) Anti-Chi1/Chi2 IgA antibodies. Levels of specific antibodies in groups are depicted and categorized according to the color and symbol key.

**Figure 3 vaccines-12-00726-f003:**
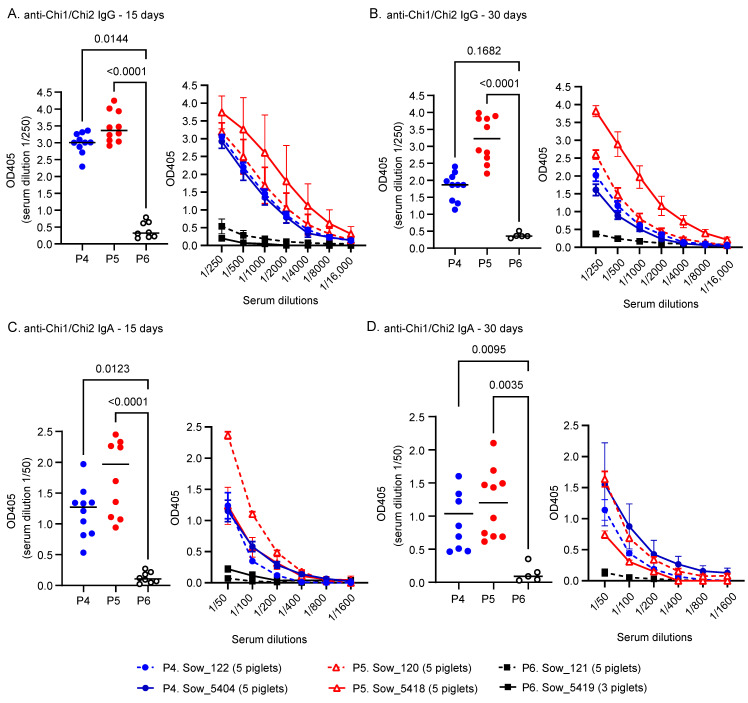
Passive transfer of maternal antibodies. Levels of specific antibodies in groups of piglets born to and nursed by each sow are depicted and categorized according to the color and symbol key. (**A**,**B**) Serum levels of specific IgG at days 15 and 30, respectively. ELISA results are shown with a serum dilution of 1/250 and with various serum dilutions. (**C**,**D**) Serum levels of specific IgA at days 15 and 30, respectively. ELISA results are shown with a serum dilution of 1/50 and with various serum dilutions. The results are expressed as absorbance values at 405 nm for each serum dilution. A successful transfer of maternal antibodies from vaccinated sows to piglets was observed, showing a natural decline as piglet age increases. Statistical analysis was performed using the Kruskal–Wallis test, followed by Dunn’s multiple comparison test. *p* < 0.05 was considered significant.

**Table 1 vaccines-12-00726-t001:** Detection frequency of genes of interest in STEC strains and other *E. coli* strains isolated from swine.

	No. of Genomes	No. (%) of Genomes Harboring the Gene of Interest						
		*stx2e*	*cah*	*ompT*	*hes*	*cah, ompT,* or *hes*	*agn43*	*fedA*
STEC	172	167 (97.1) *	60 (34.8) *	82 (47.6) *	3 (1.7) *	106 (63.5) *	27 (15.6)	46 (26.7) *
Other *E. coli*	601	0	26 (4.3)	143 (23.8)	0	152 (25.3)	74 (12.3)	0

* *p* < 0.05 compared with other *E. coli* isolated from swine via two-tailed Fisher’s exact test.

## Data Availability

The data that support the findings of this study are available on request from the corresponding author, RMV. Amino acid sequences of chimeric proteins are not publicly available due to legal restrictions and an ongoing international patent application.

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
