# Peer review of "Evaluation of the Humoral Response after Immunization with a Chimeric Subunit Vaccine against Shiga Toxin-Producing Escherichia coli in Pregnant Sows and Their Offspring"

_vaccines, 2024, doi:10.3390/vaccines12070726_

Round 1
Reviewer 1 Report
Comments and Suggestions for Authors
The manuscript entitled “Evaluation of the humoral response after immunization with a chimeric subunit vaccine against Shiga toxin-producing Escherichia coli in pregnant sows and their offspring” represents an important contribution to the topic addressed, which is highly topical.
The focus on Cah, OmpT, and Hes proteins is reasonable and based on sound science although they cannot be considered the only ones worthy of interest for the purpose of setting up potential broad-spectrum vaccines against the countless strains of STEC circulating in pig herds and beyond.
Although the authors explain the reason for their choice, it would be interesting if they elaborated in the introduction and later in the discussion on why it is potentially more effective to choose these vaccine targets rather than others, especially with regard to antibody response in terms of IgA and IgAs production, based on the currently available scientific literature.
On the other hand, I find no objection to the experimental plan and methodology adopted and described in detail in the "materials and methods" section, for which I congratulate the Authors.
I believe that the in-depth study requested is a more than sufficient supplement to the manuscript, which I hope will be published sooner rather than later.
Author Response
Reviewer 1.
The manuscript entitled “Evaluation of the humoral response after immunization with a chimeric subunit vaccine against Shiga toxin-producing Escherichia coli in pregnant sows and their offspring” represents an important contribution to the topic addressed, which is highly topical.
The focus on Cah, OmpT, and Hes proteins is reasonable and based on sound science although they cannot be considered the only ones worthy of interest for the purpose of setting up potential broad-spectrum vaccines against the countless strains of STEC circulating in pig herds and beyond.
Although the authors explain the reason for their choice, it would be interesting if they elaborated in the introduction and later in the discussion on why it is potentially more effective to choose these vaccine targets rather than others, especially with regard to antibody response in terms of IgA and IgAs production, based on the currently available scientific literature.
On the other hand, I find no objection to the experimental plan and methodology adopted and described in detail in the "materials and methods" section, for which I congratulate the Authors.
I believe that the in-depth study requested is a more than sufficient supplement to the manuscript, which I hope will be published sooner rather than later.
Response:
Thank you for your comments regarding the selection of Cah, OmpT, and Hes proteins as vaccine targets. We appreciate your suggestion to elaborate on why these particular antigens were chosen over others, particularly in relation to their potential to induce IgG and IgA antibody responses.
First, our choice of Cah, OmpT, and Hes as vaccine targets is grounded in their documented immunogenicity and prevalence in STEC strains isolated from humans and cattle. Importantly, this study shows that Cah and OmpT are also widely distributed among STEC strains isolated from swine. Moreover, these proteins have shown considerable conservation across diverse STEC strains, suggesting their potential to provide broad-spectrum protection.
In ungulates, including cattle and swine, IgG constitutes over 75% of all immunoglobulins in colostrum, emphasizing its pivotal role in neonatal immunity [1]. The epitheliochorial placenta in sows restricts the transfer of maternal antibodies during gestation, rendering piglets dependent on lactogenic immunity for early protection [2]. In particular, neonatal piglets absorb these colostral antibodies, primarily IgG, non-selectively through the gut during the first hours of life. However, this ability to absorb immunoglobulins significantly diminishes between 18 and 36 hours after birth, a period known as "gut closure" [3,4].
Additionally, functionally intact IgG within the intestinal lumen can bind antigens, contributing to immune exclusion [1,5–9]. This dual function of IgG, providing both systemic and mucosal immunity, underscores the importance of developing vaccines capable of inducing responses from both IgG and IgA.
In the context of the immune responses elicited by our vaccine candidate, it is important to note that IgG was the predominant antibody type generated, with lesser amounts of IgA (Figure 2). In particular, the P5 group showed a faster humoral response after the first dose, which could be related to a higher concentration of the chimeric proteins. The slight increase in specific IgA levels in the control group after the second dose might suggest non-specific immune responses or exposure to STEC during the study. Although no STEC was isolated from feces, initial PCR results indicated the presence of this bacterium in the herd.
This vaccine candidate increased the levels of specific IgG in the serum of pregnant sows and, consequently, in the colostrum, leveraging the natural physiological mechanisms of immune protection in ungulates. An interesting finding was the effective passive transfer of specific antibodies from immunized sows to their offspring, with antibody levels remaining relatively high from 15 to 30 days post-birth. However, a slight decrease was observed, reflecting a progressive decline in maternal protection as the piglets aged. The long-term persistence of these specific antibodies in piglets is uncertain, and further active immunization may be necessary.
Although the serum levels of IgA in vaccinated sows were not notably high compared to those in control sows (Figure 2), it is noteworthy that piglets nursed by immunized sows exhibited a significant increase in serum IgA (Figure 3). This result indicates that, along with IgG, IgA was also effectively transferred passively to the piglets.
This information was included on lines 335 – 386.
1. Hurley, W.L.; Theil, P.K. Perspectives on Immunoglobulins in Colostrum and Milk. Nutrients 2011, 3, 442–474, doi:10.3390/nu3040442.
2. Wagstrom, E.A.; Yoon, K.J.; Zimmerman, J.J. Immune Components in Porcine Mammary Secretions. Viral Immunol. 2000, 13, 383–397, doi:10.1089/08828240050144699.
3. Staley, T.E.; Bush, L.J. Receptor Mechanisms of the Neonatal Intestine and Their Relationship to Immunoglobulin Absorption and Disease. J. Dairy Sci. 1985, 68, 184–205, doi:10.3168/jds.S0022-0302(85)80812-2.
4. Everaert, N.; Van Cruchten, S.; Weström, B.; Bailey, M.; Van Ginneken, C.; Thymann, T.; Pieper, R. A Review on Early Gut Maturation and Colonization in Pigs, Including Biological and Dietary Factors Affecting Gut Homeostasis. Anim. Feed Sci. Technol. 2017, 233, 89–103, doi:10.1016/j.anifeedsci.2017.06.011.
5. Human Neonatal Fc Receptor Mediates Transport of IgG into Luminal Secretions for Delivery of Antigens to Mucosal Dendritic Cells.
6. Kobayashi K, Ogata H, Morikawa M, Iijima S, Harada N, Yoshida T, Brown WR, Inoue N, Hamada Y, Ishii H, Watanabe M, H.T. Distribution and Partial Characterisation of IgG Fc. Gut 2002, 51, 169–177.
7. Maaser, C.; Housley, M.P.; Iimura, M.; Smith, J.R.; Vallance, B. a; Finlay, B.B.; Schreiber, J.R.; Varki, N.M.; Kagnoff, M.F.; Eckmann, L. Clearance of Citrobacter Rodentium Requires B Cells but Not Secretory Immunoglobulin A (IgA) or IgM Antibodies. Infect. Immun. 2004, 72, 3315–3324, doi:10.1128/IAI.72.6.3315-3324.2004.
8. Kamada, N.; Sakamoto, K.; Seo, S.-U.; Zeng, M.Y.; Kim, Y.-G.; Cascalho, M.; Vallance, B.A.; Puente, J.L.; Núñez, G. Humoral Immunity in the Gut Selectively Targets Phenotypically Virulent Attaching-and-Effacing Bacteria for Intraluminal Elimination. Cell Host Microbe 2015, 17, 617–627, doi:10.1016/j.chom.2015.04.001.
9. Montero, D.A.; Garcia-Betancourt, R.; Vidal, R.M.; Velasco, J.; Palacios, P.A.; Schneider, D.; Vega, C.; Gómez, L.; Montecinos, H.; Soto-Shara, R.; et al. A Chimeric Protein-Based Vaccine Elicits a Strong IgG Antibody Response and Confers Partial Protection against Shiga Toxin-Producing Escherichia Coli in Mice. Front. Immunol. 2023, 14, 1–17, doi:10.3389/fimmu.2023.1186368.

Reviewer 2 Report
Comments and Suggestions for Authors
vaccines-3003350
Roberto M. Vidal and their colleagues presented a continued study aiming at an effective and safe vaccine against STEC, to safeguard food security and food safety. The overall study is conclusive and convincing, however, there are still issues to be properly addressed.
Abstract:
Line 36-37. I do not think only antibodies test could ensure protection effectiveness, without a challenge assay. Please turn down what you mean.
Line 41 "cross species vaccination", overestimate.
Introducation:
Line 78-79.
There are many key VFs in E.coli and many fimbrial adhesins are involved, including fim (J Bacteriol. 1994 Apr;176(8):2227-34). Additionally, a similar bug with Salmonella FimH adhesin has shown a robust role in FimH allelic variation targeting different host (Nat Commun. 2015 Oct 30;6:8754), resembling the way like toxin, which could be incorporated in the above section (Line 70-76).
MM:
Line 130, please clarify why these gene and their epitope were used.
Considering the challenge assay was not conducted, I would add a future direction or limitation part.
I am also very interested in those protective antibodies, and a correlation between the overall serum from sow and piglets, which is very interested to explore.
A few additional references could be added to the discussion or introduction to increase the readership and general interest for the protein way for developing vaccine, somewhat similar like Reverse Vaccinology (Vaccines (Basel) . 2023 Apr 18;11(4):865), and some new evidence for pig or pork associated ETEC.
World J Microbiol Biotechnol. 2023 Apr 28;39(7):174
Sci Rep. 2023 Feb 24;13(1):3247
Int J Food Microbiol. 2023 Apr 16;391-393:110134
Biomolecules. 2023 Nov 30;13(12):1726
Comments on the Quality of English LanguageIn general, with good language, but additional polishing will further improve the manuscript.
Author Response
Reviewer 2
Roberto M. Vidal and their colleagues presented a continued study aiming at an effective and safe vaccine against STEC, to safeguard food security and food safety. The overall study is conclusive and convincing, however, there are still issues to be properly addressed.
Abstract:
- Line 36-37. I do not think only antibodies test could ensure protection effectiveness, without a challenge assay. Please turn down what you mean.
Response:
We agree with the reviewer that a challenge assay is necessary to demonstrate the vaccine's efficacy. This study focused on evaluating the vaccine candidate's immunogenicity by measuring levels of specific IgG and IgA antibodies. While these results are promising indicators of vaccine effectiveness, we recognize that they do not conclusively prove protection against infection.
We modified the summary to clarify this point: “These humoral responses demonstrate the immunogenicity of the vaccine candidate and are preliminary indicators of its potential efficacy. However, further research is needed to conclusively evaluate its impact on STEC colonization and shedding.”
- Line 41 "cross species vaccination", overestimate.
Response:
Regarding the use of the term "cross-species vaccination" on line 41, the intention was to highlight the vaccine's potential applicability across different animal species based on previous immunogenicity studies in cows and the current findings in swine
We made the following modification to clarify this point: This study highlights the potential of maternal vaccination to protect piglets from ED and contributes to developing vaccination strategies to reduce the prevalence of STEC in various animal reservoirs.
Introduction:
Line 78-79. Many key VFs in E. coli and fimbrial adhesins are involved, including fim (J Bacteriol. 1994 Apr;176(8):2227-34). Additionally, a similar bug with Salmonella FimH adhesin has shown a robust role in FimH allelic variation targeting different hosts (Nat Commun. 2015 Oct 30;6:8754), resembling the manner like a toxin, which could be incorporated in the above section (Line 70-76).
Response:
Thank you for your insightful feedback on highlighting the diversity of fimbrial adhesins in E. coli and their role in host specificity.
In response to your suggestion, we have included the following information on lines 78-80: “Interestingly, the diversity of fimbrial adhesins found in E. coli and other enterobacteria plays a role in determining the specific hosts and tissues they can colonize and infect”. This sentence includes the references suggested by the reviewer.
Materials and methods:
Line 130, please clarify why these gene and their epitope were used.
Response:
We have included the following information on lines 140 -146:
“The Hes, OmpT, and Cah proteins are the primary targets of our STEC vaccine candidate, developed in previous studies [32,33]. We included the stx2e gene to confirm that the STEC strains analyzed were pathogenic to swine. Additionally, the agn43 gene was included to ensure the specific detection of the cah gene rather than its homologs. Given the importance of the F18ab fimbria in existing vaccines against STEC in pigs, we included the fedA gene to compare the protective coverage provided by our target proteins against established vaccine antigens.
- Considering the challenge assay was not conducted, I would add a future direction or limitation part. I am also very interested in those protective antibodies, and a correlation between the overall serum from sow and piglets, which is very interested to explore.
Response:
In response to your suggestion, we have included the following information on lines 392 - 400: “While this study provides preliminary data on the immunogenicity of our vaccine candidate in pigs, it does not include a challenge assay to assess protection against infection directly. Future research will include challenge assays to evaluate the effect of the vaccine on STEC colonization and shedding in pigs. Additionally, we will explore the correlation between serum antibody levels in sows and their piglets to better understand the transfer and efficacy of induced immunity.”
- A few additional references could be added to the discussion or introduction to increase the readership and general interest …
Response:
We thank the reviewer for the valuable suggestion regarding including additional bibliographic references. We have carefully considered the references and incorporated two of them into our manuscript to enrich the content further.
- Haque, M.; Wang, B.; Mvuyekure, A.L.; Chaves, B.D. Growth Behavior of Shiga Toxin-Producing Escherichia Coli, Salmonella, and Generic E. Coli in Raw Pork Considering Background Microbiota at 10, 25, and 40 °C. Int. J. Food Microbiol. 2023, 391–393, 110134, doi:10.1016/j.ijfoodmicro.2023.110134.
- Zhang, G.; Fu, Y.; Li, Y.; Li, Q.; Wang, S.; Shi, H. Oral Immunization with Attenuated Salmonella Choleraesuis Expressing the FedF Antigens Protects Mice against the Shiga-Toxin-Producing Escherichia Coli Challenge. Biomolecules 2023, 13, doi:10.3390/biom13121726
Please refer to lines 51 - 53 in the introduction and 336 - 338 in the discussion, where these references have been integrated to support our analysis and broaden the context of our findings. We believe these additions significantly enhance our study's readership and general interest, and we appreciate the opportunity to refine our manuscript with this critical input.

Reviewer 3 Report
Comments and Suggestions for Authors
The study aimed to investigate the efficacy of a novel vaccine against coliform infections in pigs.
The major criticism in this study is the very small number of animals employed in the study. Really, only three animals per vaccinated group is not a sufficient number of animals to test the efficacy of a vaccine. Consequently to that, the statistical methods employed are wrong. The authors should redo the analysis by using techniques specific for small numbers.
Otherwise, the work was well-executed and the manuscript is clearly written.
In view of the above, I suggest that the authors shorten the manuscript significantly to align it with the small number of animals used in their work, make clear that these are preliminary and limited results and resubmit the manuscript as a communication.
Author Response
Reviewer 3
The major criticism in this study is the very small number of animals employed in the study. Really, only three animals per vaccinated group is not a sufficient number of animals to test the efficacy of a vaccine. Consequently, to that, the statistical methods employed are wrong. The authors should redo the analysis by using techniques specific for small numbers.
Otherwise, the work was well-executed and the manuscript is clearly written.
In view of the above, I suggest that the authors shorten the manuscript significantly to align it with the small number of animals used in their work, make clear that these are preliminary and limited results and resubmit the manuscript as a communication.
Response:
We appreciate the reviewer's critical assessment of the sample size used in our study and the statistical methods employed. We acknowledge that conducting experiments with a small number of animals, particularly only three per vaccinated group, is less than ideal for robust statistical analysis and may limit the generalizability of our findings. Using pregnant sows in a biosecure facility adds further complexity due to the size, cost, and ethical considerations associated with these animals.
However, it is important to note that while the number of pregnant sows was limited, our study also included a substantial number of their piglets. Consistent with the above, we employed non-parametric statistical analysis, specifically the Kruskal-Wallis test, to analyze the serum-specific antibody levels in these piglets, which supports the reliability of our results despite the small sample size of sows.
In recognition of the limitations posed by the small sow sample size, we have carefully adjusted our manuscript to emphasize the preliminary nature of our findings. We believe that the revised statistical analysis and the additional modifications address the concerns raised and clarify the impact of our research findings.

Round 2
Reviewer 3 Report
Comments and Suggestions for Authors
The authors have addressed all the issues raised previously and have improved the manuscript. I have no further comments.
Author Response
On behalf of all the authors, I would like to thank the Reviewer 3 for their advice and suggestions on the revision of our manuscript " Evaluation of the humoral response after immunization with a chimeric subunit vaccine against Shiga toxin-producing Escherichia coli in pregnant sows and their offspring" by Vidal R et al. Thank the reviewer for the insightful comments and careful review of our manuscript, which have significantly increased its quality.